# Exploring the Influence of YouTube on Digital Health Literacy and Health Exercise Intentions: The Role of Parasocial Relationships

**DOI:** 10.3390/bs14040282

**Published:** 2024-03-28

**Authors:** Jongho Kim, Heeok Youm, Sujin Kim, Hongjun Choi, Dohee Kim, Sungeun Shin, Jinwook Chung

**Affiliations:** Department of Sports Science Convergence, College of Arts, Dongguk University, Seoul 04620, Republic of Korea; kikara77@snu.ac.kr (J.K.); worldladies@dongguk.ac.kr (H.Y.); 2022120544@dgu.ac.kr (S.K.); hongjun0802@dongguk.edu (H.C.); 2018122029@edu.ac.kr (D.K.); sseab90@dongguk.edu (S.S.)

**Keywords:** digital health literacy, parasocial relationships, health exercise behavior intention, YouTube

## Abstract

The purpose of this study was to analyze the mediating role of digital health literacy and the moderating effect of parasocial relationships on the relationship between the viewing experience of health exercise-related YouTube content and the intention for health exercise behavior. Based on the health action process approach, this study established a foundational theoretical model to analyze how digital health literacy mediates the impact of media viewing experience on health exercise behavior intention. Additionally, this study examined the moderating effect of parasocial relationships with YouTube creators. For empirical analysis, variables were measured using a self-administration method among 409 randomly sampled consumers of YouTube health exercise content. The collected data were analyzed using a structural equation model incorporating mediation parameters, and a multigroup model analysis was conducted to understand differences based on parasocial relationships. The results revealed that increased YouTube viewing experience enhanced cognitive, skill, and evaluative components of digital health literacy, which were significant factors in increasing health exercise behavior intention. Notably, the mediating effect of cognition played a crucial role, and the strengthening effect of parasocial relationships on this relationship was confirmed. These findings can be utilized as practical foundational data for designing digital health communication strategies, particularly in developing motivational mechanisms that encourage consumers to engage voluntarily and consistently in health behaviors based on online health information.

## 1. Introduction

The development of digital convergence and media platforms is improving individuals’ quality of life. In today’s society, people can effortlessly access necessary information online anytime and anywhere. This advancement in media technology has significantly impacted personal health management, fitness training, and grooming appearance [1]. In contrast to the past, when health-related knowledge was obtained via professionals or specialized books, in the present, online access to such information is readily available [2]. Online health information, known for its quick update frequency and ease of access, differs from traditional methods of acquiring information, as it often enables direct interaction with the creators of the content [3]. The heightened interest in health, especially following the COVID-19 pandemic, and the preference for non-contact exercises like home training have led to an exponential increase in demand for online exercise prescriptions and health fitness coaching videos [4].

Previously, there were significant barriers to access for health- and exercise-related information and training, such as the need for direct delivery from professionals or interpreting specialized books [5]. Now, numerous experts post their expertise in the form of web content, and videos that condense information from specialized books are easily accessible [5]. Several studies have provided evidence that the ability to search for, understand, and use digital health literacy is related to actual health status and health behavior intentions [1,5,6]. On YouTube, various creators post content about home training, weight training, and exercises and diets for weight loss, acting as influencers who disseminate relevant information to a broad online audience [7,8].

However, the vast amount of information available online can sometimes hinder consumers’ search for and selection of information and may not always lead to actual behavior. For example, content that has been indiscriminately posted on multiple platforms can create barriers to finding and selecting the desired content, especially when it is not well categorized. In [9], the authors suggest that the cognition of acquired information itself can motivate behavior, while [10] asserts that sustained health actions occur when the information is evaluated and trusted. Moreover, Ref. [8] emphasizes that for digital health literacy to lead to consistent exercise behavior, strong parasocial relationships, such as trust and homophily with content providers, are necessary. These studies highlight that digital media facilitates the acquisition of health literacy, which is then utilized in health behavior and exercise participation, underscoring the importance of considering third variables that influence psychological mechanisms depending on the context.

Therefore, this study aims to ascertain the impact of exposure to online health exercise information and the formation of digital health literacy on the intention for actual health exercise behavior. Specifically, we analyze the relationship between digital health literacy formed through YouTube watching and the intention to continuously participate in health exercises. In addition, this study examines the moderating effect of perceived parasocial relationships with YouTube creators. This research is significant in that it provides practical foundational data for developing digital health communication strategies and encouraging consumers to engage voluntarily and continuously in health behaviors based on online health information. It also holds practical significance in improving the quality of life of the wider society. Academically, by analyzing the impact of digital health literacy and parasocial relationships on health behavior intentions, this study expands the theoretical framework describing the relationship between online health information consumption and health behavior changes.

Overall, the import of this research extends beyond providing empirical evidence; it suggests a paradigm shift towards empowering individuals to actively engage in their health management through digitally accessible information. By examining how digital health literacy and parasocial relationships impact exercise intentions, our work contributes to the expanding body of knowledge in behavioral science. It offers insights into the mechanisms through which digital engagement can influence health behaviors, particularly in the context of long-term health management. This exploration is crucial for understanding how to leverage digital platforms effectively in promoting healthy behaviors, underscoring the potential of such interventions to improve the quality of life and facilitate the management of chronic conditions. Ultimately, our findings seek to inform the development of nuanced, digitally informed interventions that harness the potential of online platforms to foster sustained engagement in health-promoting behaviors. By integrating the dynamics of digital health literacy and parasocial interactions, this study illuminates pathways for enhancing behavioral strategies aimed at the long-term management of health conditions, thereby contributing to the broader goal of promoting public health and wellbeing through informed, interactive digital communication.

## 2. Theoretical Background

### 2.1. Influence of Online Health Exercise Content Experience on Behavioral Intentions

Traditionally, mass media has played a role in promoting the positive functions of health exercises through campaigns, changing public attitudes and beliefs about health and exercise, and urging action [11]. As times have changed, with the shift from mass media to digital media as the mainstream, modern individuals are enhancing their health beliefs and intentions by actively seeking information online [12]. In this era of digital media, people obtain information through various channels, choosing and consuming content at their convenience to change their health beliefs and attitudes [12]. In particular, social media users, like those on YouTube, are not passive recipients but rather actively use media to satisfy their needs [13], and viewers involved in watching YouTube exercise content are increasing their interest in and participation intention for exercises through media consumption [14]. Numerous studies using various models have theoretically proven the relationship between digital media consumption and exercise participation intentions. For example, applying the health empowerment theory, Ref. [15] explained how individuals gain psychological motivation through personal/social resources and perform active health-promoting behaviors. Similarly, Ref. [16] described the series of processes leading from motivation to volition and then to behavior in the health action process approach (HAPA).

The HAPA model, among various theoretical models explaining changes in health exercise behavioral intentions according to media consumption experiences, has been increasingly used to describe how digital media consumers change their beliefs and attitudes toward health exercises, leading to behavioral intention [17]. This model provides a theoretical framework for explaining an individual’s health behavior process and is highly regarded for understanding the discrepancy between intentions and behavior regarding exercise [16,18]. The HAPA model distinguishes between the motivational stage that leads to the intention for behavior, the volitional stage that leads to actual behavior, and the action stage, explaining the series of processes in adopting and maintaining health behaviors like exercise [19,20].

Specifically, the motivational stage forms the intention for exercise through various paths, where one’s perception of their health or the expectation of positive benefits from a specific behavior arises [20]. Generally, self-efficacy and risk perception acquired from past experiences play an important role in generating behavioral motivation. The volitional stage, on the other hand, involves planning behavior and predicting outcomes, playing a direct role in actual behavior [20]. Personal commitment to planning and consistently maintaining health behaviors like exercise involves various factors in each stage [20]. According to [18], information and education provided by the media involve both the motivational and volitional stages, with media persuasion and invitation messages involving the motivational stage and scientific knowledge and empirical information involving the volitional stage. Moreover, Ref. [21] emphasized that empowerment, formed through social support and acquiring knowledge and skills, plays a crucial role in behavior change. Studies [15,22] have reported that online health information behavior plays an important role in the empowerment needed for individuals to carry out health behaviors.

As can be seen, past research trends have revealed that knowledge and indirect experiences acquired via media use play a significant role in motivating and planning health exercise behaviors. Hence, we can hypothesize that as engagement in watching YouTube content related to exercise increases, the intention to participate in exercise strengthens. However, despite high interest in and intention to exercise among YouTube exercise content viewers, it cannot be assumed that they are regularly performing exercises, indicating a need to analyze the factors that determine behavior change and maintenance besides motivational factors like intention. Many prior studies have assumed personal intentions as the most direct antecedent in predicting exercise behavior, but the issue has been raised that intention does not always lead to behavior [23,24]. With these points in mind, this study focuses on the mediating role of health literacy between watching engagement and behavioral intention. The following section presents theoretical discussions related to the role of health literacy.

**Hypothesis** **1.**
*The extent of the watching experience, characterized by both time spent and level of engagement with online health exercise-related content, is positively associated with an increased intention to participate in health exercise behaviors.*


### 2.2. The Mediating Role of Digital Health Literacy

Health literacy refers to an individual’s capacity to acquire, understand, and evaluate health-related information and make health-related decisions, leading to appropriate action [25]. Particularly, health literacy can be interpreted as a personal resource, involving active resource-acquiring behaviors such as assessing one’s situation, obtaining necessary information, and acting upon it to acquire health resources [26]. Generally, the greater the quantity and quality of information available, the more resources one can acquire.

In the context of health literacy, the digital media age offers easy access to a vast array of information, facilitating the active acquisition of health resources [26]. However, acquiring incorrect information or misjudgments can lead to a loss of health resources, underscoring the need for correct health literacy accumulation [27]. Digital health literacy, a concept expanded from health literacy, involves the ability to search for, interpret, and apply necessary health-related information among the wealth that is available online, and it is gaining attention as a crucial concept in the digital media environment [27]. Digital health literacy can be acquired through various channels like social media, online communities, digital education, mobile health apps, and blogs. The authors of [28] highlighted that easily accessible digital media like YouTube plays a vital role in enhancing society’s health literacy levels. In particular, the concept of health literacy is essential for individual health management and promotion and is highly significant from a public health perspective, making digital health literacy as acquired through digital channels socially important [25].

Digital health literacy comprises three domains: cognitive, skill, and evaluation [25,27]. Each element plays a crucial role in health behavior, especially in sustaining the intention to participate in health exercise behavior [26,27]. The cognitive aspect involves understanding and interpreting health-related information, which is essential for planning and applying methods suited to one’s situation. Skill refers to the ability to effectively find and use health-related information, including searching capabilities and applying them in daily life. Evaluation involves the capacity to judge the reliability and usefulness of the acquired information, and it involves assessing the credibility of the source, the accuracy of the information, and its appropriateness.

Numerous studies using theories such as the theory of planned behavior explain that positive attitudes toward health behavior, subjective norms, and perceived behavioral control enhance health exercise behavior intentions [29,30]. In the same vein, individuals with high health literacy are likely to strongly recognize the importance and necessity of health exercise, the skills to plan and execute it, and the ability to apply the right information [28,31]. Building on this foundation, it becomes evident that active engagement with media, such as watching health-related content on YouTube, plays a pivotal role in the formation of digital health literacy [28]. This process is instrumental in enhancing the cognitive, skill, and evaluative dimensions of digital health literacy [28]. Specifically, viewers engaging with YouTube content benefit from a direct enhancement of their cognitive abilities, as they are exposed to a wide range of health-related information that aids in the understanding and internalization of complex health concepts and practices. Furthermore, this active engagement facilitates the development of practical skills for seeking, evaluating, and applying health information in real-life scenarios. Lastly, the evaluative component of digital health literacy is strengthened as viewers learn to critically assess the credibility and utility of the information provided, enabling them to make informed decisions about their health behaviors. Thus, the interactive nature of digital platforms like YouTube serves not only as a source of information but also as a dynamic environment for developing comprehensive digital health literacy.

We posit that the abundance of information, high accessibility, and interactivity of online media platforms like YouTube positively influence the formation of the cognitive, skill, and evaluation aspects of digital health literacy [27,28]. In the theoretical foundation of our study, we delve into the intricate dynamics of health literacy, specifically focusing on its three fundamental components: cognition, skill, and evaluation. Traditionally, these elements have been collectively examined within the broader construct of health literacy. However, our research adopts a distinct approach by treating each component as an independent mediator. This decision is anchored in a nuanced understanding that, although they are interconnected, cognition, skill, and evaluation each play a unique role in the process of translating digital media engagement into health behavioral intentions.

**Hypothesis** **2.**
*The extent of the watching experience, characterized by both time spent and level of engagement with online health exercise-related content, is positively associated with an enhancement in the cognitive aspect of digital health literacy.*


**Hypothesis** **3.**
*The extent of the watching experience, characterized by both time spent and level of engagement with online health exercise-related content, is positively associated with an improvement in the skill aspect of digital health literacy.*


**Hypothesis** **4.**
*The extent of the watching experience, characterized by both time spent and level of engagement with online health exercise-related content, is positively linked to an increase in the evaluative aspect of digital health literacy.*


Numerous studies have empirically demonstrated the critical role of digital health literacy formation in health behavior [26,27], with recent research applying various theories to predict behaviors like exercise [32]. The authors of [32] revealed that individual health literacy plays a significant and positive role in exercise participation among preventive health exercise behaviors through interaction with self-efficacy. In [33], the authors found that adolescents participating in sports club activities have lower health literacy levels compared to non-participating students and that health literacy significantly influences participation volume. The authors of [34] provide multifaceted evidence that digital media spaces are suitable for teaching health literacy that is appropriate for sports or health exercise behavior. Further, it has been argued that digital literacy education is particularly effective for adolescents compared to traditional literacy education. Thus, many studies have empirically proven the significant correlation between digital health literacy formation and physical activity participation and have highlighted the importance of audiovisual elements in digital media, which is easily accessible to the public, in enhancing health literacy.

In a similar vein, numerous studies have confirmed the important role of digital health literacy in strengthening health exercise behavior intentions in modern society. That is, individuals with high digital health literacy can effectively find, evaluate, and apply health information provided online, and it is predicted that such individuals make more evidence-based health decisions and engage in sustained health behaviors. Therefore, this study predicts that the experience of watching health exercise-related content enhances digital health literacy, subsequently strengthening health exercise behavior intentions. This suggests that individuals can internalize health information obtained through digital media, transforming it into the motivation needed for adopting and maintaining healthy lifestyles, and that understanding this relationship can play a vital role in building a healthy society. Moreover, in the current media consumption environment, where the utilization of online health information is becoming widespread, there is a high risk of information disparity and gaps for those with low health literacy, despite substantial online accessibility. Therefore, understanding the relationship between information and health literacy acquired from free OTT platforms like YouTube and health exercise behavior intentions is vital from a public health promotion perspective.

**Hypothesis** **5.**
*An enhancement in the cognitive aspect of digital health literacy is positively associated with an increased intention to participate in health exercise behaviors.*


**Hypothesis** **6.**
*An improvement in the skill aspect of digital health literacy is positively associated with an increased intention to participate in health exercise behaviors.*


**Hypothesis** **7.**
*An increase in the evaluative aspect of digital health literacy is positively linked to an increased intention to participate in health exercise behaviors.*


### 2.3. The Moderating Effect of Parasocial Relationships

On the other hand, the intensity of the audience’s experience and psychological state changes according to various execution factors of media content [35]. According to [36]’s SMCR (sender, message, channel, receiver) model, which explains the relationship between media viewers’ experiences and their behavioral intentions or motivations, content senders (sender) must effectively deliver appropriate information (message) using useful channels (channel) to change or reinforce consumers’ (receiver) thoughts. In other words, from the message receiver’s perspective, not only the message content but also who the sender is plays a crucial role in the response to the message. The message source effect model, specifically the credibility model, reveals that factors like the speaker’s credibility, attractiveness, and familiarity influence engagement with the message, leading to changes in memory and attitude. Recent studies have validated this understanding by examining parasocial relationships [37,38,39].

Parasocial relationships refer to the imaginary human relationships formed between media audiences and media figures [40,41]. This concept reflects the idea that the public interacts socially with figures in media, explaining emotional bonds through the perceived familiarity, trust, attractiveness, and credibility of the speaker [40,41]. Specifically, this can be explained through imaginary and experiential human relationships. Imaginary relationships refer to psychological bonds formed through interactions with media figures, while experiential relationships refer to psychological bonds formed through actual dialogue or interactions [40,41]. Previous studies report that when a strong parasocial relationship is formed, the audience exhibits high recognition rates for the information delivered by the speaker, changing attitudes or beliefs in response to its high trust in the message [7,37,42,43]. In health communication, changing beliefs or attitudes about health through media to transform them into actual health behaviors is set as an important task. From this perspective, parasocial relationships formed through media are predicted to play a significant role as a moderating variable in the relationship between digital health literacy formation through media engagement and behavioral intentions.

Specifically, among related research examples, Ref. [7] found that parasocial relationships strengthen the motivation of users who follow fitness influencers on YouTube, with those who are already participating in fitness exercises showing enhanced behavior, while non-participants only experienced strengthened information-acquisition motivation. Similarly, Ref. [44] reported that the formation of parasocial relationships based on the perceived physical attractiveness, social attractiveness, and similarity of trainers in YouTube home training positively influences the continuous intention to exercise. Further, Ref. [43] demonstrated that the social attractiveness, physical attractiveness, task attractiveness, and content quality of fitness influencers enhance parasocial relationships between influencers and viewers. The authors of [38] presented results indicating that the strength of parasocial relationships based on the perceived similarity, attractiveness, and credibility of social media influencers enhances the persuasiveness of the information they present. In [37] the authors reported that in public service ads related to obesity, the formation of parasocial relationships based on the perceived likability, trust, and competence of the speaker enhances exercise participation efficacy.

Thus, numerous previous studies illuminate the positive impact of the extent of parasocial relationships with health and health exercise content deliverers on changing attitudes and motivations, tending to focus on content recipients like YouTube users. Modern media consumers actively seek and adopt the information they want and tend to strengthen behavioral intentions or motivations following media consumption. Correspondingly, information delivered by speakers with highly formed parasocial relationships is predicted to further enhance the tendency of increased behavioral intentions following watching. Consequently, this study additionally hypothesizes that parasocial relationships enhance the relationship between digital health literacy formation and continuous exercise behavior intentions following YouTube health exercise content watching.

**Hypothesis** **8.**
*The relationship between online health exercise watching experience, health exercise behavior, and digital health literacy differs between groups with high versus low parasocial relationships with the information deliverer.*


## 3. Methodology

### 3.1. Research Hypotheses and Model

This study establishes research hypotheses and a model based on the theoretical background previously discussed to analyze the relationship between health exercise information experienced on YouTube and its impact on actual health exercise behavior through the formation of digital health literacy. Specifically, utilizing self-determination theory and the theory of planned behavior, we formulated hypotheses regarding the relationship between YouTube watching experience and health exercise behavior intention, the mediating role of health literacy as a mediating variable, and the moderating effect of parasocial relationships. Figure 1 presents the research model according to the hypotheses of this study.

### 3.2. Research Subjects

For this study, using random sampling, we conducted a survey on a nationwide sample of 409 consumers who consumed health exercise-related YouTube content. The survey period was from 1 August to 15 August 2023, a total of two weeks. After the survey concluded, 41 insincere responses were excluded, leaving a total of 368 responses for use in this study. The demographic characteristics of the research subjects are presented in Table 1. To investigate the experience of consuming health exercise-related YouTube content, preliminary items were constructed, and respondents who did not subscribe to health-information-related channels were all excluded in the first phase and did not participate in the survey.

According to the demographic characteristics of the research subjects, in terms of gender, there were 191 males (51.90%) and 177 females (48.09%). In terms of age distribution, the 30 s group was the largest with 101 individuals (27.45%), followed by the 20 s with 86 (23.37%), the 40 s with 73 (19.84%), and the teens with 54 (14.67%). Geographically, 182 participants (49.46%) were from the capital area, 98 (26.63%) from metropolitan cities, and 88 (23.91%) from other regional areas. In terms of educational level, college graduates were the most numerous with 169 individuals (45.92%), followed by college students at 79 (21.47%), graduate school students at 62 (16.85%), graduate school graduates at 24 (6.52%), and middle/high school students at 54 (14.67%).

### 3.3. Measurement and Data Analysis

The research subjects responded to the questionnaire using the self-administration method. The measured variables were watching experience, digital health literacy, intention for health exercise behavior, and parasocial relationships, with all items except watching time measured on a 7-point Likert scale (1 = strongly disagree, 7 = strongly agree). The survey was conducted online, utilizing a web-based platform where participants were first presented with a detailed overview of the research objectives and ethical considerations. Prior to accessing the survey questions, participants were required to go through a consent procedure. This involved reading a consent form that outlined the study’s purpose, the voluntary nature of their participation, the confidentiality of their responses, and their right to withdraw at any time. Consent was obtained by participants selecting a ‘consent’ button, thereby affirming their agreement to partake in the study under the outlined terms. This investigation was rigorously designed and executed in strict observance of the ethical guidelines delineated in the Declaration of Helsinki. The Institutional Review Board, recognizing the adherence to these ethical standards, granted its formal approval. This endorsement certifies that the research complied with the necessary ethical norms and standards for studies involving human participants, emphasizing our commitment to ethical research practices throughout the execution of the study. To mitigate nonresponse bias, we employed strategies such as follow-up contacts and demographic comparisons between respondents and non-respondents, ensuring a representative sample. Furthermore, we addressed common method variance by implementing procedural remedies like anonymity and reverse-coded items, alongside statistical checks using a single-factor test, confirming the minimal impact of CMV on our findings.

For watching experience, we refer to the method proposed by [45,46], i.e., measuring both watching time and engagement and then integrating them into a combined index. Watching time was measured in minutes using log data from the YouTube application’s personal settings screen for a week and multiplying it by the percentage of watched health exercise content. Watching time, calculated as a continuous variable, was rank ordered and then recoded into an interval scale on a 7-point scale for analysis with other variables. Watching engagement measured the degree of exposure and interest in related media content by counting the number of subscribed channels related to health and exercise on the YouTube application. This variable was also standardized and recoded into an interval scale on a 7-point scale.

Regarding digital health literacy, this study utilizes the measurement items for digital health literacy as proposed by [47]. Digital health literacy is composed of three factors—cognitive, skill, and evaluation—each consisting of three items. Specifically, the three cognitive items are “I am aware of the various information related to health exercise online” (DHL_cognitive-1), “I know how to find health exercise-related information online” (DHL_cognitive-2), and “I know how to find the health exercise-related information I need online” (DHL_cognitive-3). The skill items include “I can find the answers I need for my health exercise among various online information” (DHL_skill-1), “I know exactly how to use the health exercise information I found online” (DHL_skill-2), and “I have the skills needed to assess the accuracy and quality of the health exercise information I found online” (DHL_skill-3). The evaluation items are “I have the ability to recognize false or low-quality health exercise-related information” (DHL_evaluation-1), “I can evaluate the truthfulness or quality of health exercise-related information that I have been exposed to or experienced at least once” (DHL_evaluation-2), and “I have the evidence needed to make decisions when using health exercise-related information obtained online” (DHL_evaluation-3).

To measure the intention for health exercise behavior variable, this study adapts and the intention to exercise scale proposed in [48]. Specifically, intention for health exercise behavior comprised three items: “I will engage in health exercise for my health” (Behavior 1), “I think engaging in exercise for my health is very important” (Behavior 2), and “I will continue to engage regularly in health exercise in the future” (Behavior 3).

For parasocial relationships, we measured viewers’ imaginary and experiential human relationships with creators appearing in YouTube health exercise-related content by adapting items from [40,49] and others. Specifically, the items were “I think I have a lot in common with the YouTube creator of the health exercise content I primarily watch”, “The YouTube creator of the health exercise content I primarily watch is friendly and familiar”, and “I think I could have a comfortable conversation with the YouTube creator of the health exercise content I primarily watch”. For use as a moderating variable, we calculated the average value of these three items and then divided groups into high and low on the basis of this criterion.

We analyzed the collected data using the statistical package programs STATA 15 and AMOS 18. Specifically, we conducted basic analysis and variable management using STATA 15 and structural equation modeling, mediating effect analysis (effect decomposition analysis), and moderating effect analysis (multigroup analysis) using AMOS 18. The demographic characteristics of the research subjects were verified through frequency analysis. In addition, confirmatory factor analysis and Cronbach’s alpha coefficient were used to check convergent validity, discriminant validity, and internal consistency among items.

## 4. Result

### 4.1. Descriptive Statistics, Factor Correlation, and Reliability Analysis

Table 2 presents the descriptive statistics of the main variables measured in this study. All measured variables were continuous, and their mean, standard deviation, minimum, maximum, skewness, and kurtosis values were examined. Regarding the mean values and standard deviations of the constructs, the mean watching experience in terms of watching time was found to be 3.71 (SD = 2.22), and that of watching engagement was 3.29 (SD = 1.59). The average watching experience, composed of watching time and watching engagement, was 3.50 with a standard deviation of 1.90. For digital health literacy, cognitive was 4.91 (SD = 1.46), followed by skill at 4.36 (SD = 1.191) and evaluation at 4.02 (SD = 1.75). In the case of parasocial relationships, the average was 3.38 (SD = 1.51), and when divided on the basis of the average value, the high group consisted of 169 individuals and the low group of 199 individuals. The average for health exercise behavior intention was 5.17 (SD = 2.62). Additionally, to verify the normality of the main variables, skewness and kurtosis values were checked. Skewness ranged from −0.74 to 1.03, and kurtosis from 1.52 to 3.38, indicating that they satisfied normality [50].

Table 3 presents the correlation analysis results for the main variables. In the process of analyzing the structural equation model, performing a correlation analysis is a crucial preliminary step. It explores the linear relationships between the measurement variables included in the model and is necessary before establishing direct or indirect pathways between them. The results in Table 3 identify the interrelations among the main variables measured in this study using Pearson correlation coefficients. The correlation coefficients between all variables being 0.691 or lower suggests the presence of moderate relationships between the variables. This implies that while the variables are related, the strength of these relationships is not very high.

Furthermore, when the correlation between predictor variables is high, the problem of multicollinearity, which can lead to inaccurate estimations in regression analysis, can arise. According to [51], multicollinearity should be suspected when the correlation coefficient exceeds 0.80. In this study, all variables show correlation coefficients of 0.691 or lower, which indicates the absence of multicollinearity, suggesting that the variables provide independent information. In summary, the correlation analysis reported in Table 3 demonstrates appropriate linear relationships between the variables and, simultaneously, a low risk of multicollinearity, indicating that the data structure is suitable for statistical modeling.

### 4.2. Reliability and Validity of Measurement Tools: Confirmatory Factor Analysis

To verify the reliability and validity of the measurement tools, we conducted Cronbach’s alpha testing and confirmatory factor analysis (CFA). The reliability and validity of measurement tools can vary depending on the target group, so this study not only examined the entire group but also divided and analyzed groups according to the degree of parasocial relationship with YouTube health exercise content creators. The average value for parasocial relationship attitude was 3.38, and groups were formed on the basis of this mean value (mean split). The results are presented in Table 4.

The confirmatory factor analysis results for the measurement model showed good fit for the entire group (χ^2^ = 198.822 [df = 105, *p* < 0.01], χ^2^/df = 1.893, Tucker–Lewis Index [TLI] = 0.927, comparative fit index [CFI] = 0.944, root-mean-square error of approximation [RMSEA] = 0.058, standardized root-mean-square residual [SRMR] = 0.062), the group with a high level of parasocial relationship (χ^2^ = 211.454 (df = 105, *p* < 0.01), χ^2^/df = 2.013, TLI = 0.897, CFI = 0.881, RMSEA = 0.062, SRMR = 0.069), and the group with a low level of parasocial relationship (χ^2^ = 231.778 (df = 105, *p* < 0.01), χ^2^/df = 2.207, TLI = 0.847, CFI = 0.834, RMSEA = 0.064, SRMR = 0.067).

The reliability analysis showed that the Cronbach’s alpha values for each subdomain in the entire group ranged from 0.782 to 0.830, in the high par-asocial relationship group from 0.803 to 0.895, and in the low group from 0.800 to 0.890. Generally, a reliability level above 0.6 is considered good, indicating a trustworthy internal reliability of the items used in this study. The composite reliability (CR) values for the convergent validity of the measurement tools were 0.711–0.896 for the entire group, 0.760–0.910 for the group with a high level of parasocial relationship, and 0.720–0.880 for the low group. These values meet the acceptable criteria of 0.5 [52].

Additionally, the average variance extracted (AVE) values for discriminant validity were 0.501–0.859 for the entire group, 0.500–0.880 for the high parasocial relationship group, and 0.560–0.840 for the low group. These values, which are within the acceptable range of 0.5–0.95, confirm that the measurement tools used in this study also have adequate discriminant validity [53]. Overall, these results confirm that the measurement tools used in this study have high internal reliability as well as convergent and discriminant validity. This satisfies the conditions for the measurement tools that are needed to verify the multigroup structural equation model and hypotheses presented earlier, enhancing the validity and generalizability of the research findings.

### 4.3. Structural Equation Modeling Analysis and Mediation Effect Verification

In this study, a structural equation model with mediating variables was developed and analyzed to explore the relationship between the experience of watching health exercise-related content and the formation of digital health literacy, as well as its subsequent effect on health exercise behavioral intention. Specifically, health literacy was composed of three subfactors: cognitive, skill, and evaluation. These three factors, while independent, form the concept of digital health literacy and were thus set as mediating variables, with paths accordingly analyzed for each. A total of seven hypotheses (Hypothesis 1 to 7) were tested. Prior to analyzing each path and its mediating effect, the model fit of the structural equation model was assessed and confirmed to be suitable for interpretation (χ^2^(df) = 479.08(219), TLI = 0.915, CFI = 0.903, RMSEA = 0.056) [54].

This study’s structural equation model examined the impact of watching experience on the three main components of digital health literacy (cognitive, skill, evaluation) and how these components affect the intention for health exercise behavior. According to the model testing results in Table 5 and Figure 2, watching experience had a statistically significant positive effect on health exercise behavioral intention (b = 0.26, *p* < 0.05, SE = 0.110), supporting Hypothesis 1. Watching experience positively influenced the cognitive aspect of digital health literacy (b = 0.83, *p* < 0.001, SE = 0.051), supporting Hypothesis 2 and suggesting that cognitive ability can be considerably enhanced by watching experience. Similarly, watching experience significantly and positively affected the skill (b = 0.93, *p* < 0.001, SE = 0.072) and evaluation (b = 0.86, *p* < 0.001, SE = 0.045) aspects of digital health literacy, supporting Hypotheses 2 and 3. Furthermore, all three components of digital health literacy—cognitive (b = 0.93, *p* < 0.01, SE = 0.11), skill (b = 0.32, *p* < 0.01, SE = 0.088), and evaluation (b = 0.24, *p* < 0.01, SE = 0.093)—had a positive impact on health exercise behavioral intention, supporting Hypotheses 4, 5, and 6. Therefore, it can be confirmed that digital health literacy partially mediates the effect of watching experience on health exercise behavioral intention, as revealed through direct effect analysis [55].

Additionally, to statistically confirm the mediating effects of the three components of digital health literacy—cognitive, skill, and evaluation—we executed an indirect effects verification using bootstrap, as suggested by Preacher and Hayes [56]. The indirect effect of digital health literacy in mediating the impact of watching experience on intention for health exercise behavior was confirmed through significance testing of the indirect confidence intervals (95%) using bootstrapping. The analysis results are presented in Table 6.

According to the results, the direct effect of watching experience was statistically significant on intention for health exercise behavior (β = 0.26, *p* < 0.05). The indirect effects of the three components of digital health literacy—cognitive (β = 2.121), skill (β = 0.836), and evaluation (β = 0.585)—were all statistically significant. Notably, the indirect effect of the cognitive aspect was especially high, underscoring the importance of digital health literacy in influencing the intention for ongoing participation in health exercise. The indirect confidence intervals (95%) using bootstrapping did not include 0, confirming the statistical significance of these indirect effects. The relatively high indirect effect of the cognitive aspect of digital health literacy suggests that understanding and awareness of health information play a significant role in influencing the intention for ongoing participation in health exercise. This implies that information gained through watching experience, when appropriately recognized and interpreted, can lead to a sustained intention to engage in health exercise. The indirect effects of the skill and evaluation aspects were significant but relatively lower compared to the cognitive, indicating that the ability to understand and interpret health information may be more critical in forming the intention for ongoing participation in health exercise than technical skills or information evaluation abilities.

### 4.4. Moderating Effect of Parasocial Relationships: Multigroup Structural Equation Analysis

To investigate whether the relationship between the experience of watching YouTube content related to health exercise and its effect on health exercise behavioral intention through digital health literacy differs depending on the level of parasocial relationship with YouTube creators, we divided parasocial relationships into high and low groups on the basis of the average value and conducted a multigroup structural equation analysis. The average value of parasocial relationships was 3.38, and among the 368 participants, 169 were in the high parasocial relationship group, while 199 were in the low group.

Specifically, to compare the path coefficients between the two groups, we analyzed the similarity of the two models by comparing the constrained and unconstrained models, focusing on form equivalence and structural equivalence. The results are presented in Table 7. The unconstrained model, or the form equivalence model, showed a good fit (χ^2^ = 455.822 (df = 174), CFI = 0.903, RMSEA = 0.049). The measurement equivalence model, which constrained the paths between the latent and measured variables to be equal showed a relatively good fit of the model and data (χ^2^ = 590.112 (df = 194), CFI = 0.912, RMSEA = 0.047) but was not statistically significantly different from the form equivalence model. This confirms that the observed variables measuring each construct were perceived identically across groups. The structural equivalence model, which constrained the variances and covariances of latent variables also showed a relatively good fit (χ^2^ = 601.331 (df = 210), CFI = 0.911, RMSEA = 0.043).

The statistically significant difference between the form and measurement equivalence models confirms that the groups divided by generation have an effect as a moderating variable. Accordingly, we conducted a comparative analysis of the path coefficients between the groups with high versus low parasocial relationships. The results are shown in Table 8. Specifically, the impact of watching health exercise-related YouTube content on intention for health exercise behavior was found to be significant in both the high (b = 0.304, *p* < 0.001) and low (b = 0.211, *p* < 0.01) parasocial relationship groups, and the difference in path coefficients between the groups was also significant. The high group experienced a statistically larger impact compared to the low group (χ^2^= 0.381, *p* < 0.01).

Moreover, the impact of the cognitive ability of digital health literacy on intention for health exercise behavior was significant in both the high (b = 0.873, *p* < 0.001) and low (b = 0.500, *p* < 0.01) groups, and the difference in path coefficients was also significant (χ^2^ = 0.301, *p* < 0.05). Furthermore, the impact of the skill ability of digital health literacy on intention for health exercise behavior was significant in the high group (b = 0.448, *p* < 0.05) but not in the low group (b = 0.255, *p* > 0.05), with a significant difference in path coefficients between the groups (χ^2^ = 0.412, *p* < 0.01). No significant differences in path coefficients between groups were found for the other paths. Consequently, the moderating effect of parasocial relationships was found to strengthen the relationship between watching experience and intention for health exercise behavior as well as the impact of the cognitive and skill abilities of digital health literacy on intention for health exercise behavior intention.

## 5. Conclusions

In the field of health exercise, health communication through digital platforms like YouTube plays a crucial role in promoting individual and societal health by disseminating health information and encouraging a healthy lifestyle through smart healthcare, exercise, and sports [57]. This study comprehensively explores the impact of digital health literacy formed through the consumption of health exercise content on YouTube on the intention to engage in health exercise. Specifically, we aimed to understand how digital health literacy formed through YouTube viewing influences actual health exercise behavior intention and to contribute to the improvement of the quality of life of society at large through the design of effective digital health communication strategies and development of mechanisms for motivating voluntary and continuous health behavior.

In addition, we conducted a multigroup analysis to understand the differences in the cognition and behavior of digital media content consumers according to the parasocial relationships they form with information providers. Specifically, we tested seven hypotheses (H1–H7) to analyze the relationship between YouTube health exercise content viewing experience, digital health literacy, and intention for health exercise behavior, as well as an additional hypothesis (H8) to understand the differences based on the level of parasocial relationships. The major findings and discussion are as follows.

## 6. Discussion

Firstly, the direct effect of the experience of viewing media content related to health exercise on the intention to engage in health exercise was found to be significant. This suggests that digital media like YouTube provides an important platform for accessing and consuming health information and that the enhancement of knowledge and perception through these platforms promotes changes in health behavior. The authors of [13,58] have indicated that the consumption of health information through digital media influences individuals’ health beliefs and attitudes. While these studies suggest that the consumption of information can influence psychological factors, they have not clearly elucidated the direct impact on actual behavioral intentions. However, [59] reported that the information acquired from social media can directly change consumer behavior quantitatively and qualitatively and emphasized the need for research on related mechanisms.

With these points in mind, this study fills the gap in previous research, demonstrating empirically that the consumption of information through digital media can directly influence health behavioral intentions, making a significant academic contribution. Additionally, this study empirically validates the potential rise in individual health exercise behavioral intention as assumed by numerous studies that have applied the health empowerment theory and health action process approach. Our results also provide significant practical implications, offering substantial support for campaigns and strategy development related to health behavior enhancement using digital media and underscoring the necessity of health communication development using YouTube media.

Secondly, the impact of the viewing experience of health exercise media content on the subfactors of digital health literacy—namely, cognition, skill, and evaluation—was found to be significant, with the order of impact magnitude being skill, evaluation, and then cognition. This indicates that consumers can enhance their ability to search for, interpret, and evaluate the quality of information through online content consumption, emphasizing the role and importance of digital media, which has been overlooked in health literacy research. Specifically, this study provides new theoretical insights into the role and importance of digital media in health literacy research. Extending the health literacy models proposed by [60,61], this study specifically illuminates the impact of health exercise content consumption through online platforms like YouTube on the formation of digital health literacy.

Moreover, this study goes beyond the digital health literacy concept proposed by [47] and makes a significant theoretical contribution to the realm of health communication in terms of the utilization of online platforms like YouTube. Indeed, our results represent an important theoretical advancement in the field of health literacy research. When [60,61] proposed their health literacy models, they primarily focused on traditional communication channels. However, this study analyzes the influence of the media consumption experience in the digital age, particularly on platforms like YouTube, on health literacy formation. This research deepens the digital health literacy concept proposed by [47], suggesting a new direction in health communication research using online media.

Previous research in digital health literacy somewhat heuristically conceptualized it as a single concept [28] or predominantly emphasized cognitive elements [47]. This study overcomes the limitations of previous studies by providing a more detailed and systematic understanding of health literacy formation, which extends beyond just providing cognitive information and emphasizes campaigns related to the exploration and identification of information in communication strategies.

In terms of practical and policy implications, these results offer important guidelines for health exercise promotion communicators and information providers in establishing strategies for providing health information using digital media, especially YouTube. For instance, public health and exercise participation campaigns and physical education programs can be designed to enhance the public’s health decision-making capabilities and promote health behaviors by utilizing YouTube. This recommendation is based on this study’s findings that enhancing the ability to evaluate and apply information is crucial. For example, agencies like the Centers for Disease Control and Prevention (CDC) in the United States that aim to promote health exercise participation can use YouTube to enhance the public’s health decision-making capabilities and promote health behaviors. Furthermore, this study’s results suggest that it is important to provide cognitive information and give policy weight to campaigns that are related to the exploration and identification of information in communication strategies.

Third, the impacts of the cognitive, skill, and evaluation aspects of digital health literacy on intention for health exercise behavior were all significant, with the influence being the greatest for the cognitive aspect. This suggests that cognitive elements play a crucial role in enhancing actual behavioral intention in terms of digital health literacy. Health literacy in the cognitive domain includes the ability to understand and interpret health-related information, which is essential for correctly perceiving health information and applying it to one’s health situation. This emphasis on the correct interpretation and application of health information in forming health behavioral intention is also highlighted in [26]. Overall, the prominence of the cognitive domain in impacting the intention for health exercise behavior indicates that understanding and interpreting health information plays a decisive role in leading to actual behavior change. Further, it suggests that accurately interpreting health information is the first step in forming health behavior intention and that the subsequent technical and evaluative elements complement and enhance this cognitive understanding.

Moreover, although the impact of YouTube watching experience on technical ability was greater than that on cognitive ability, the analysis indicates that cognitive ability is more crucial for inducing actual behavioral intention. This implies that a comprehensive and systematic approach beyond mere information accessibility and technical ability is needed for health education through media to lead to actual health behavior intention. Digital media like YouTube increases the accessibility of health information and enhances related technical skills, but the importance of cognitive elements should not be overlooked for these factors to translate into actual health behavior intention.

Cognitive ability encompasses the understanding of the interpretation, processing, and application of information, which is essential for users when applying health information to their actual situations. Therefore, health information providers need to consider strategies that go beyond the mere provision of information to enable users to understand this information and convert it into health behavior intention. For instance, YouTube content should not only provide information but also practical guides or examples of how this information can be applied in real life. Including educational content that strengthens cognitive elements to enable users to interpret and apply health information to their situations is crucial. Such an approach can contribute to transitioning from the mere acquisition of knowledge to the formation and maintenance of healthy lifestyle habits. Health-related media content creators and educators need to develop strategies that foster more effective health behavior changes through such a multidimensional approach. This also signifies an important direction in the development of digital health literacy that advocates a user-centered educational approach to foster continuous improvement in health behavior.

Fourth, the relationship between health exercise watching experience and health exercise behavioral intention was partially mediated by the cognitive, skill, and evaluation capabilities of digital health literacy. This result supports the findings of [1,5], which suggest that health behavior is enhanced by the health literacy acquired from media experience. However, unlike previous studies which focused on general health-related behavior, this research centers on proactive health promotion and preventive activities, specifically, health exercise behavior. This distinction highlights the academic significance of this study. Unlike general health behavior, health exercise behavior requires active effort and participation, where the digital media’s role in information acquisition and interpretive skills is crucial.

Specifically, this study’s findings demonstrate that digital health literacy plays a significant role in forming health exercise behavioral intention, particularly highlighting the greatest indirect effect of cognitive ability. This suggests that knowledge acquired through the digital consumption of health exercise information is a key element that leads to active health behavior changes. In particular, these results provide a basis for devising innovative strategies to strengthen the relationship between digital health literacy and health exercise behavioral intention. Generally, health exercise promotion campaigns should enhance interaction with the public and provide tailored information using AI and data analytics. Recent health exercise promotion communication strategies have been expanding message reach and impact through collaboration with social media influencers and are transforming information consumption into an enjoyable and participatory activity through gamification and storytelling. Policymakers can leverage these new strategies to establish support mechanisms and explore ways to enhance public digital health literacy through educational programs and workshops. Such approaches can contribute not only to individual health behavior changes but also to forming a health culture and promoting public health at a societal level.

Fifth, the parasocial relationships that media consumers perceive with YouTube health exercise content creators were found to have a moderating effect, enhancing the mediating effect of digital health literacy. This result supports the findings of [7,37,38] that strong parasocial relationships with media figures positively impact information reception and behavior change. Past research in digital health literacy has focused more on the type and delivery of information and the psychological state of the receiver rather than on the attributes of the information provider such as attractiveness, credibility, and trustworthiness. However, this study, considering the specificities of the online digital media consumption environment, measures the relationship between the receiver and the information provider as well as the receiver’s perception of this relationship via parasocial relationships, and it investigates its impact on the acquisition of digital health literacy and health exercise behavioral intention.

Ultimately, the results indicate that the positive effect of health exercise watching experience and the formation of digital health literacy on behavioral intention was stronger in groups with highly formed parasocial relationships. This new perspective suggests that the relationship between the information provider and the receiver plays a significant role in the reception of health information and behavior change. Particularly in the digital media environment, a parasocial relationship acts as a key element in the receiver trusting the health information more and actively applying it to their life. This relationship is a crucial motivational factor that encourages the receiver beyond the mere reception of information to internalizing it and transitioning to actual health behavior.

Leveraging these results, health information providers and media creators can adopt strategies to strengthen parasocial relationships in health content creation. Influencers, for instance, can share their experiences and actively interact with users to build deeper trust and bonds. Communication involves a speaker and a listener, and the relationship between the two is a significant topic in educational settings. Content creators that provide health exercise information can be considered educators from the viewers’ perspective, and the relationship between the two is a crucial variable for educational effectiveness. In today’s digital content educational environment, building relationships between information providers and receivers is emerging as an essential element for the effective delivery and reception of health information. This will provide new directions for the design and implementation of health communication strategies and lay an important foundation for health behavior changes and public health promotion in the digital age.

## 7. Limitations and Future Research Suggestions

This study has examined the relationship between the consumption of health exercise content on YouTube, digital health literacy, and the intention to engage in health exercise behaviors, ultimately underscoring the significance of digital media in the field of health communication and providing practical strategies for promoting a healthy lifestyle. Nevertheless, we highlight important limitations and directions for future research, as outlined below.

Firstly, a key limitation is this study’s failure to systematically control for the role of third variables considering theoretical models of health behavior. This suggests a limitation in fully understanding the complex interactions between variables when interpreting the results. Although this study was confined to specific theoretical models, future research should more intricately explore the multidimensional aspects of health behavior by integrating various theoretical perspectives or developing new theoretical models. Specifically, by considering a wide range of psychological and environmental factors that influence health behavior, more effective information can be extracted. For instance, a multilayered analysis based on theoretical models of health behavior could explore the interactions between individual psychological traits, social environments, and media message characteristics to provide essential foundational data for devising effective health communication strategies.

Secondly, this study did not consider the specificity of health literacy tools. Specifically, given the absence of health literacy measurement tools that are optimized for the context of health exercise behavior, we adopted tools suggested in previous research and verified their reliability and validity. However, to ensure completeness, it is necessary to use tools in consideration of the specificity of the context. In other words, future research should develop digital health literacy measurement tools that are specific to health exercise behavior. Such tools can measure various aspects of health literacy more accurately and contribute to a better understanding of behavior changes specific to health exercise.

Developing tools reflecting this specificity would also facilitate a more detailed and accurate evaluation of particularly important aspects of health exercise behavior. For example, tools that pertain to understanding specific exercise methods, setting health goals, self-health management capabilities, and critically evaluating online health information can reflect the unique requirements of health exercise behavior. This will provide essential data for the design and evaluation of health exercise promotion programs and contribute to developing personalized strategies for individual health improvement. Moreover, developing tools considering macro-level specificity will also contribute significantly to health communication research by enabling a deeper understanding of what information health-information consumers actually need and how they process and use that information. Such tools can also enable more accurate assessments of the impact of various health communication channels and strategies on specific health behaviors, contributing to the development of more effective health communication strategies.

Our methodological approach, focused on dissecting the distinct roles of cognition, skill, and evaluation, exposes a pivotal limitation—the potential oversight of their collective impact. The intricacies of these components’ interrelations remain unexplored within our analysis, suggesting that the synergistic effects integral to health literacy might be concealed by our current research design. This acknowledgment underlines the need for subsequent studies to embrace comprehensive models that incorporate the interactivity and causal interconnections among these mediators. Such research endeavors would not only close existing gaps but also enrich our grasp of the concerted influence these elements exert on health behavior in digital settings, ultimately guiding the evolution of more nuanced health literacy frameworks and communication strategies.

Additionally, the present study acknowledges that demographic differences, such as age and gender, may lead to variability in YouTube viewing behaviors, motivations, satisfaction, and subsequent actions. Although our model differentiated groups based on parasocial relationships, acknowledging demographic characteristics remains a noted limitation. While efforts were made to control demographic variables by employing a sample with a wide age range and balanced gender composition, potential biases inherent in such a design cannot be fully disregarded. To enhance the validity of future research, it is essential to broaden the scope of study participants and conduct subgroup analyses using foundational demographic variables. This approach would allow for more robust and reliable research findings by examining the differential effects across diverse groups, thus addressing the nuances of audience heterogeneity that were not fully explored in the current study Building on this, future studies should aim to enhance methodological rigor, particularly by adopting a broad sampling strategy that encompasses various population groups and regions to increase representativeness and the overall scope of the research. This will increase the generalizability of the research findings and provide a more comprehensive understanding of the health behaviors and health literacy of individuals from diverse social and cultural backgrounds. Further, to reduce the subjective bias in research that relies on survey methods, various data collection methods such as behavioral observations, in-depth interviews, and focus group discussions can be implemented. This mixed-methodological approach will overcome the limitations of subjective self-reported data and provide deeper and more multifaceted insights. Finally, to overcome the limitations of a cross-sectional research design, a longitudinal research design can be adopted. This approach will allow for a clearer understanding of the causal relationships by tracking how health behaviors and health literacy change over time. Longitudinal studies are also important for evaluating the long-term effects of health communication strategies.

Theoretical and methodological improvements to the research design will contribute to providing more reliable and empirical results in the fields of health communication and digital health literacy. Further, they will lay the foundation for the effective communication of health information, fostering changes in individual health behaviors, enhancing public health, and building a healthier society. Advancements in such research will strengthen the theoretical foundation of health communication academically and practically; inspire new directions for the design, implementation, and evaluation of health communication strategies; and contribute to the development of public health promotion strategies that are suitable for the digital age of health information.

## Figures and Tables

**Figure 1 behavsci-14-00282-f001:**
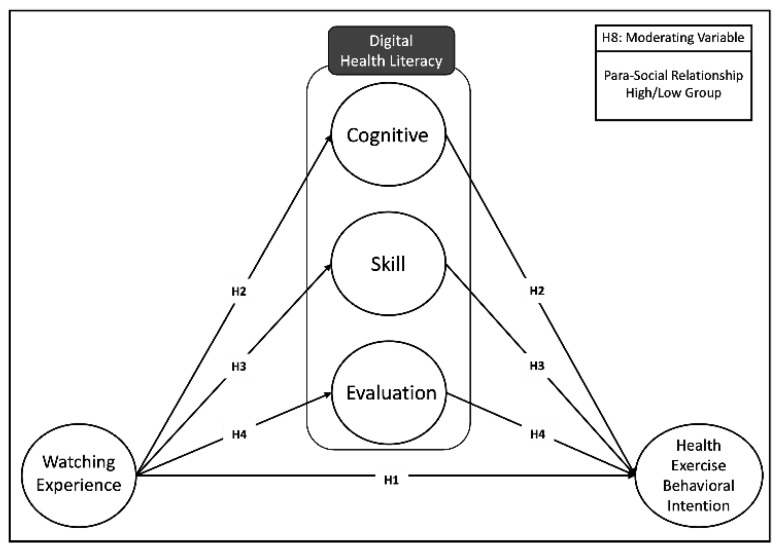
Research hypotheses and model: conceptual model outlining the hypothesized pathways influencing digital health literacy.

**Figure 2 behavsci-14-00282-f002:**
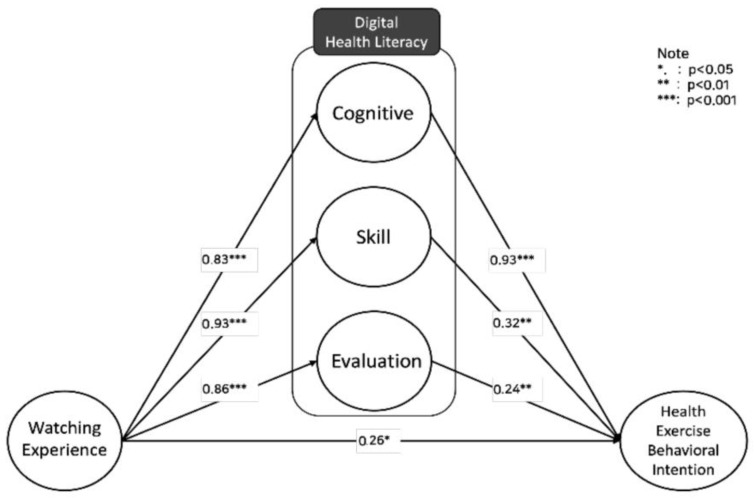
Research model testing result: diagram displaying the standardized path coefficients for the relationships affecting digital health literacy.

**Table 1 behavsci-14-00282-t001:** Demographic characteristics.

Variable	*n*	%
* **Gender** *		
Male	191	51.90
Female	177	48.10
* **Purpose for use** *		
Weight control	139	64.95
Healthy lifestyle	83	8.97
Physical fitness	146	26.09
* **Age** *		
10 s	54	14.67
20 s	86	23.37
30 s	101	27.45
40 s	73	19.84
50 s and above	54	14.67
* **Region** *		
Capital area (Seoul/Gyeonggi)	182	49.46
Metropolitan cities	98	26.63
Other (by region)	88	23.91
* **Education level** *		
Middle/high school student	54	14.67
College student	79	21.47
College graduate	169	45.92
Graduate school student	62	16.85
Graduate school graduate	24	6.52

**Table 2 behavsci-14-00282-t002:** Descriptive statistics.

Variable	*M*	*SD*	Skewness	Kurtosis
Watching experience				
Time	3.71	2.22	−0.03	1.52
Engagement	3.29	1.59	0.59	1.65
Digital health literacy				
Cognitive	4.91	1.46	−0.74	2.80
Skill	4.36	1.91	−0.38	2.03
Evaluation	4.02	1.75	−0.03	2.13
Parasocial relationships	3.38	1.51	1.03	3.38
Intention for health exercise behavior	5.17	2.62	−0.67	2.80

*Note.* Min. and max. values for all variables were 1 and 7, respectively.

**Table 3 behavsci-14-00282-t003:** Correlation between independent variables.

Variable	1	2	3	4	5	6
1. Watching experience	1.000					
2. Cognitive	0.673 ***	1.000				
3. Skill	0.679 ***	0.652 ***	1.000			
4. Evaluation	0.691 ***	0.623 ***	0.692 ***	1.000		
5. Parasocial relationship	0.697 ***	0.622 ***	0.639 ***	0.683 ***	1.000	
6. Intention for health exercise behavior	0.419 ***	0.336 ***	0.324 ***	0.584 ***	0.442 ***	1.000

*** *p* < 0.001.

**Table 4 behavsci-14-00282-t004:** Confirmatory factor analysis and item reliability verification.

	Item	b	S.E	t	β	α			AVE			C.R		
						Total	H	L	Total	H	L	Total	H	L
Watching experience	Time	1	-	-	0.636	0.782	0.803	0.800	0.501	0.500	0.560	0.711	0.760	0.720
	Engagement	0.913	0.183	4.97	0.577									
Cognitive	C 1	1	-	-	0.835	0.830	0.840	0.835	0.812	0.830	0.790	0.867	0.880	0.850
	C 2	1.106	0.083	13.29	0.926									
	C 3	0.993	0.094	10.56	0.808									
Skill	S 1	1	-	-	0.772	0.805	0.815	0.810	0.800	0.820	0.780	0.862	0.870	0.840
	S 2	1.078	0.114	9.40	0.824									
	S 3	0.923	0.107	8.61	0.793									
Evaluation	E 1	1	-	-	0.662	0.805	0.815	0.810	0.792	0.810	0.770	0.858	0.865	0.835
	E 2	1.274	0.175	7.28	0.854									
	E 3	1.280	0.174	7.34	0.897									
Health exercise behavior intention	HEBI 1	1	-	-	0.880	0.885	0.895	0.890	0.859	0.880	0.840	0.896	0.910	0.880
	HEBI 2	0.902	0.085	10.60	0.781									
	HEBI 3	0.974	0.069	14.08	0.915									

Note. AVE = average variance extracted; CR = composite reliability. Total model fit: χ^2^(104) = 222.235, RMSEA = 0.051, CFI = 0.944, TLI = 0.932, SRMR = 0.048; high parasocial relationship group (N = 169): χ^2^(104), RMSEA = 0.066, CFI = 0.927, TLI = 0.916, SRMR = 0.054; low parasocial relationship group (N = 199): χ^2^(104), RMSEA = 0.064, CFI = 0.930, TLI = 0.919, SRMR = 0.052.

**Table 5 behavsci-14-00282-t005:** Structural equation model analysis: direct effect path coefficients.

Hypothesis	Path	b	*SE*	CR	Judge
H1	Watching Experience → Health Exercise Behavior Intention	0.26 *	0.110	2.20	Accept
H2	Watching Experience → Cognitive	0.83 ***	0.051	18.22	Accept
H3	Watching Experience → Skill	0.93 ***	0.072	19.01	Accept
H4	Watching Experience → Evaluation	0.86 ***	0.045	18.51	Accept
H5	Cognitive → Intention for Health Exercise Behavior	0.93 ***	0.11	17.09	Accept
H6	Skill → Intention for Health Exercise Behavior	0.32 **	0.088	10.88	Accept
H7	Evaluation → Intention for Health Exercise Behavior	0.24 **	0.093	8.12	Accept

Note. CR = composite reliability. * *p* < 0.05. ** *p* < 0.01. *** *p* < 0.001.

**Table 6 behavsci-14-00282-t006:** Decomposition analysis table for structural model effects.

Path	Direct Effect	Digital Health Literacy Indirect Effect			Total Effect	CIs (Bias-Corrected, 95%)
		Cognitive	Skill	Evaluation		
Watching Experience → Intention for Health Exercise Behavior	0.26	2.121	0.836	0.585	3.802	0.010–0.061 ***

Note. CI = confidence interval. *** *p* < 0.001.

**Table 7 behavsci-14-00282-t007:** Model fit comparison between unconstrained and constrained models.

Model	χ^2^	*df*	CFI	RMSEA	Δχ^2^	Sig.
Form equivalence	222.235	104	0.899	0.051		
Measurement equivalence	275.342	125	0.901	0.049	25	0.069 (Reject)
Structural equivalence	301.614	146	0.900	0.050	9	0.021 (Accept)

Note. CFI = comparative fit index, RMSEA = root-mean-square error of approximation.

**Table 8 behavsci-14-00282-t008:** Comparison of path coefficient estimates between groups.

Path	∆χ^2^	Parasocial Relationships			
		High Group		Low Group	
		b	*SE*	b	*SE*
Watching Experience → Intention for Health Exercise Behavior	3.81 **	0.304 **	0.078	0.211 **	0.120
Watching Experience → Cognitive	1.12	0.978 ***	0.039	0.911 ***	0.051
Watching Experience → Skill	1.22	0.777 ***	0.048	0.897 ***	0.051
Watching Experience → Evaluation	0.93	0.944 ***	0.037	0.845 ***	0.050
Cognitive → Intention for Health Exercise Behavior	3.01 *	0.873 ***	0.169	0.500 *	0.053
Skill → Intention for Health Exercise Behavior	4.13 **	0.448 **	0.195	0.255	0.051
Evaluation → Intention for Health Exercise Behavior	0.81	0.203	0.140	0.342	0.021

* *p* < 0.05. ** *p* < 0.01. *** *p* < 0.001.

## Data Availability

The data presented in this study are available upon request from the corresponding author due to restrictions related to participant privacy concerns.

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
