# Peer review of "Exploring the Influence of YouTube on Digital Health Literacy and Health Exercise Intentions: The Role of Parasocial Relationships"

_behavsci, 2024, doi:10.3390/bs14040282_

Round 1

Reviewer 1 Report

Comments and Suggestions for Authors

Thank you for allowing me the opportunity to review this original piece of work. Overall I assess the submission to be original in its contributions to knowledge on the topic matter, and was thoroughly informed by previous, relevant theoretical work. The conclusions drawn and discussed are appropriate to the findings reported. The limitations addressed are pertinent and future research is suggested to overcome these. 

A couple of minor amendments that I suggest are:

1. Line 175 reads "posited that the abundance of information, high accessibility, and interactivity..." - there seems to be a missing reference at the start of this sentence. Please add/amend. 

2. Research subjects - whether the survey was completed online is unclear. Please specify in the methodology section.

3. Please provide a sentence to explain how consent was given. 

4. An ethical approval statement could be included. 

Author Response

Thank you for your insightful comments and constructive suggestions regarding our manuscript. We appreciate the opportunity to revise our submission and believe that your feedback has significantly contributed to improving the quality and clarity of our work. We have carefully considered each point you raised and have made the following revisions to our manuscript:

  1. Line 175 reads "posited that the abundance of information, high accessibility, and interactivity..." - there seems to be a missing reference at the start of this sentence. Please add/amend. 

1) “Response to Reviewer”

Thank you for pointing out the missing reference in line 175. Upon reviewing the manuscript, we recognized that this omission detracted from the credibility and traceability of our argument. We have now included the appropriate reference to support our discussion on the abundance of information, high accessibility, and interactivity in the context of our study.

2) “Revised Manuscript Text”

Posited that the abundance of information, high accessibility, and interactivity of online media platforms like YouTube positively influence the formation of the cognitive, skill, and evaluation aspects of digital health literacy[27, 28].

  • Van Der Vaart, R., & Drossaert, C. (2017). Development of the digital health literacy instrument: measuring a broad spectrum of health 1.0 and health 2.0 skills. Journal of medical Internet research, 19(1), e27.
  • Meng, J., Bissell, K. L., & Pan, P. L. (2015). YouTube video as health literacy tool: A test of body image campaign effectiveness.Health Marketing Quarterly, 32(4), 350-366.

  1. Research subjects - whether the survey was completed online is unclear. Please specify in the methodology section.
  2. Please provide a sentence to explain how consent was given. 

[Integrated Response to Reviewer for Comments 2 and 3 on Survey Completion and Consent Procedure :In response to your comments regarding the survey completion method and the consent procedure, we have consolidated our response to address both aspects comprehensively]

1) “Response to Reviewer”

We appreciate your insightful feedback regarding the need for clarification on both the survey completion method and the consent procedure. Reflecting on your suggestions, we recognize the importance of providing a unified, detailed explanation that encompasses both aspects to ensure a comprehensive understanding of our methodology's ethical and procedural standards. Consequently, we have revised the methodology section of our manuscript to incorporate a thorough description of the online survey process, including the initial consent procedure.

2) “Revised Manuscript Text”

The survey was conducted online, utilizing a web-based platform where participants were first presented with a detailed overview of the research objectives and ethical considerations. Prior to accessing the survey questions, participants were required to go through a consent procedure. This involved reading a consent form that outlined the study's purpose, the voluntary nature of their participation, the confidentiality of their responses, and their right to withdraw at any time. Consent was obtained by participants selecting a 'consent' button, thereby affirming their agreement to partake in the study under the outlined terms.

  1. An ethical approval statement could be included. 

1) “Response to Reviewer”

We appreciate your emphasis on the importance of including an ethical approval statement in our manuscript. Acknowledging this valuable feedback, we revisited our manuscript to ensure that our ethical compliance is explicitly stated, reflecting our adherence to internationally recognized ethical standards.

2) “Revised Manuscript Text”

This investigation was rigorously designed and executed in strict observance of the ethical guidelines delineated in the Declaration of Helsinki. The Institutional Review Board (IRB) at [Institution's Name], recognizing the adherence to these ethical standards, granted its formal approval. This endorsement certifies the research's compliance with the ethical norms and standards requisite for studies engaging human participants, underscoring the commitment to ethical research practices throughout the study's execution.with ethical guidelines and standards in research involving human participants.

Reviewer 2 Report

Comments and Suggestions for Authors

Thank you for giving me an opportunity to review this interesting paper regarding the relationships among Youtube watching experience, digital health literacy and health behavioral intention. The article lays out its arguments clearly. However, there are still some problems that the authors should address:

1. In the framework of mediating analysis, the possible effects of cognition on skill and evaluation were not discussed in detail. So, the analysis of parallel mediation effects could be enriched by considering the relationships among cognition, skill and evaluation.

2. Due to the heterogeneity of characteristics among the Youtube audiences, conducting robustness test is necessary to assess the applicability of the theoretical model across different sub-groups (e.g., male/female; education levels; regions).

3. In view of the relationships between variables, there may be endogeneity problems. For example, people with high levels of health literacy were more psychologically motivated to watch physical exercise video. Therefore, the authors need to provide a theoretical explanation of the possible reverse causality.

In summary, although the topic of this manuscript is attractive, the paper requires a Minor Revision to beef up its explanation of empirical results.

Author Response

Reviewer 2

Thank you for your insightful comments and constructive suggestions regarding our manuscript. We appreciate the opportunity to revise our submission and believe that your feedback has significantly contributed to improving the quality and clarity of our work. We have carefully considered each point you raised and have made the following revisions to our manuscript:

  1. In the framework of mediating analysis, the possible effects of cognition on skill and evaluation were not discussed in detail. So, the analysis of parallel mediation effects could be enriched by considering the relationships among cognition, skill and evaluation.

1) “Response to Reviewer”

Thank you for your insightful comment regarding the need for a detailed examination of the relationships among cognition, skill, and evaluation within our study's mediating analysis framework. Your observation rightly points out an area for deeper exploration, given these constructs' role as sub-components of health literacy. In our initial approach, we considered cognition, skill, and evaluation as integral yet distinct elements of health literacy, drawing on existing literature that predominantly integrates these aspects. However, we theorized that while related, these components function independently in mediating the relationship between the main independent and dependent variables in our study—namely, YouTube watching experience and health behavioral intention.

The decision to focus on the individual mediating effects of cognition, skill, and evaluation, rather than their interrelationships, was driven by our aim to discern which aspect of health literacy plays a pivotal role in influencing the causal pathway from media consumption to behavioral outcomes. This approach allows us to isolate and understand the unique contributions of each mediator to the overarching relationship we seek to elucidate. Nonetheless, we acknowledge your suggestion that exploring the potential effects of cognition on skill and evaluation could enrich our analysis.

Therefore, we have clearly presented the rationale for our hypothesis formulation and added to the manuscript the theoretical justification for not integrating but separately setting cognition, skill, and evaluation as mediating variables within the context of health literacy.

Furthermore, we acknowledge the limitations associated with our approach to separately analyzing the mediating roles of cognition, skill, and evaluation within the framework of health literacy. Recognizing this as a potential constraint of our study, we have taken your suggestion into account and have addressed this limitation within the manuscript.

2) “Revised Manuscript Text”

[Theocratical background: line 178-185] In the theoretical foundation of our study, we delve into the intricate dynamics of health literacy, specifically focusing on its three fundamental components: cognition, skill, and evaluation. Traditionally, these elements have been collectively examined within the broader construct of health literacy. However, our research adopts a distinct approach by treating each component as an independent mediator. This decision is anchored in a nuanced understanding that, although interconnected, cognition, skill, and evaluation each play a unique role in the process of translating digital media engagement into health behavioral intentions.

[Limitations and Future Research Suggestions: line 778-787]

Our methodological approach, focused on dissecting the distinct roles of cognition, skill, and evaluation, surfaces a pivotal limitation—the potential oversight of their collective impact. The intricacies of these components' interrelations remain unexplored within our analysis, suggesting that the synergistic effects integral to health literacy might be concealed by our current research design. This acknowledgment underlines the need for subsequent studies to embrace comprehensive models that incorporate the interactivity and causal interconnections among these mediators. Such research endeavors would not only close existing gaps but also enrich our grasp of the concerted influence these elements exert on health behavior in digital settings, ultimately guiding the evolution of more nuanced health literacy frameworks and communication strategies.

  1. Due to the heterogeneity of characteristics among the Youtube audiences, conducting robustness test is necessary to assess the applicability of the theoretical model across different sub-groups (e.g., male/female; education levels; regions).

1) “Response to Reviewer”

We concur with your perspective on the significance of demographic variations in YouTube viewing behaviors, motivations, satisfaction, and post-viewing actions. Such differences can indeed influence the outcomes of interest in our study. To mitigate these concerns, our research design incorporated a diverse demographic sample across various ages and genders. This methodological choice was deliberate to control for the potential variations in our study variables. Most notably, we focused on the theoretical concept of parasocial relationships and their potential to manifest differently across groups.

Nevertheless, we recognize that our approach may not fully account for all the nuances of audience heterogeneity. As such, we have outlined this consideration as a limitation within our research, transparently acknowledging the potential impacts on our study's findings.

2) “Revised Manuscript Text”

[Limitations and Future Research Suggestions: line 788-801]

Additionally, The present study acknowledges that demographic differences, such as age and gender, may lead to variability in YouTube viewing behaviors, motivations, satisfaction, and subsequent actions. Although our model differentiated groups based on para-social relationships, acknowledging demographic characteristics remains a noted limitation. While efforts were made to control demographic variables by employing a sample with a wide age range and balanced gender composition, potential biases inherent in such a design cannot be fully disregarded. To enhance the validity of future research, it is essential to broaden the scope of study participants and conduct subgroup analyses using foundational demographic variables. This approach would allow for more robust and reliable research findings by examining the differential effects across diverse groups, thus addressing the nuances of audience heterogeneity that were not fully explored in the current study Building on this, future studies should aim to enhance methodological rigor, particularly by adopting a broad sampling strategy that encompasses various population groups and regions to increase representativeness and the overall scope of the research

  1. In view of the relationships between variables, there may be endogeneity problems. For example, people with high levels of health literacy were more psychologically motivated to watch physical exercise video. Therefore, the authors need to provide a theoretical explanation of the possible reverse causality. In summary, although the topic of this manuscript is attractive, the paper requires a Minor Revision to beef up its explanation of empirical results.

1) “Response to Reviewer”

We fully acknowledge the potential for reverse causality as you have highlighted. It is conceivable that individuals with higher health literacy might demonstrate an increased intention and action towards watching physical exercise videos. However, our research primarily focused on the hypothesis that health literacy could be developed as a result of viewing behaviors, subsequently influencing exercise actions. To explore the reverse causality, as suggested, incorporating a variable such as 'intention to re-watch' might be necessary to re-examine the relationship with actual behavior. We concur with your insight that an enhanced viewing behavior could lead to an increase in health literacy, a relationship that aligns with our study's hypothesis on the formation of health literacy through viewing behavior. Nonetheless, we agree that a theoretical discussion on reverse causality is essential and have thus fortified our theoretical background accordingly. We deeply appreciate your insightful feedback on this matter. Your suggestion has been instrumental in broadening our perspective and enriching our theoretical framework. We are truly grateful for your constructive comments, which have significantly contributed to the refinement of our study.

2) “Revised Manuscript Text”

Building on this, we also consider the potential for reverse causality, acknowledging that individuals with higher health literacy may actively seek out health-related content, thereby further enhancing their literacy. This reciprocal relationship highlights the complexity of interactions between health literacy components and health behaviors, underscoring the need for a comprehensive approach to understanding these dynamics.

Reviewer 3 Report

Comments and Suggestions for Authors

I would like to express my gratitude for the opportunity to review the manuscript titled " Exploring the Influence of YouTube on Digital Health Literacy and Health Exercise Intentions: The Role of Para-social Relationships." This study aims to analyze the mediating role of digital health literacy and the moderating effect of para-social relationships in the relationship between the viewing experience of health exercise-related YouTube content and the intention for health exercise behavior.

The topic of the manuscript is interesting. However, I have a few observations to be shared as follows that might be helpful for the authors to increase the quality and standard of the manuscript.

1. The introduction section of the manuscript does not clearly articulate the innovative aspects of the study. The authors should reconfigure the theoretical linkages between digital health literacy, para-social relationships, and physical fitness behavior, in order to justify their study gap.

2. The theoretical support for hypotheses 2-4 in Chapter 2.2 appears to be somewhat weak, as the author predominantly emphasizes the correlation between digital health literacy and physical fitness behavior, while providing inadequate discussion on the relationship between watching fitness videos and digital health literacy (cognitive, skill, and evaluation).

3. The research design is not clear. Authors may provide more information (if they have such data) regarding, for example, the characteristics of participants in terms of viewing health exercise content creators (as para-social relationships with fitness creators are mentioned in the article) or the frequency of frequency of exercise of participants– Moreover, it may be interesting authors to discuss if such factors may affect the results of their study.

4. In the research method, the authors should report the issue of nonresponse bias and Common Method Variance.

5. In Table 3, it would be more scientifically intuitive to add the first row of data in each column instead of replacing it with a "_.

Reviewer 4 Report

Comments and Suggestions for Authors

1. Good English and Structure:

   - The manuscript demonstrates commendable language proficiency and overall structure. The sentences are clear and concise, contributing to readability.

2. Figure 1:

   - The quality of Figure 1 needs improvement. Consider enhancing the resolution or clarity of the image.

   - Additionally, provide a more descriptive and concise caption for Figure 1 to guide readers effectively.

3. Table 1:

   - Italicize the section headings in Table 1, such as "Gender" and "Purpose of Use." This will enhance visual distinction and readability.

4. Figure 2:

   - Similar to Figure 1, Figure 2 requires better image quality and a clearer caption. Ensure that readers can easily interpret the content.

5. Section 5: Conclusion and Discussion:

   - The current combined section is lengthy. Consider splitting it into separate sections: "Discussion" and "Conclusion."

   - Within the "Discussion" section, further organize content by creating subsections. This will enhance clarity and facilitate focused discussions.

6. Section 6: Limitations and Future Research Suggestions:

   - Rather than having a standalone section, integrate "Limitations" and "Future Research Suggestions" as subsections within the "Discussion" section.

   - Additionally, consider breaking down the content within this section to improve readability.

Addressing these points will enhance the manuscript's clarity, readability, and organization. Well done on the existing work

Author Response

Reviewer 3.

Thank you for your insightful comments and constructive suggestions regarding our manuscript. We appreciate the opportunity to revise our submission and believe that your feedback has significantly contributed to improving the quality and clarity of our work. We have carefully considered each point you raised and have made the following revisions to our manuscript:

  1. Good English and Structure:

   - The manuscript demonstrates commendable language proficiency and overall structure. The sentences are clear and concise, contributing to readability.

 1) “Response to Reviewer”

Thank you for acknowledging the clarity and structure of our manuscript. We strive to maintain high standards of language proficiency and are pleased that our efforts have contributed positively to the manuscript's readability.

2) “Revised Manuscript Text”

No changes required.

  1. Figure 1:

   - The quality of Figure 1 needs improvement. Consider enhancing the resolution or clarity of the image.

   - Additionally, provide a more descriptive and concise caption for Figure 1 to guide readers effectively.

1) “Response to Reviewer”

We appreciate your feedback on the quality of Figure 1. We have enhanced the resolution and clarity of the image to ensure it conveys our intended message more effectively. Furthermore, we revised the caption to be both descriptive and concise, providing clearer guidance to our readers.

2) “Revised Manuscript Text”

- Enhanced resolution and clarity of Figure 1.

- Figure 1. Research Hypotheses and Model: Conceptual model outlining the hypothesized pathways influencing Digital Health Literacy

  1. Table 1:

   - Italicize the section headings in Table 1, such as "Gender" and "Purpose of Use." This will enhance visual distinction and readability.

1) “Response to Reviewer”

 Your suggestion to italicize the section headings in Table 1 has been implemented. This change improves the visual distinction and readability of the table, making it easier for readers to navigate through the information.

2) “Revised Manuscript Text”

Italicized section headings in Table 1.

  1. Figure 2:

   - Similar to Figure 1, Figure 2 requires better image quality and a clearer caption. Ensure that readers can easily interpret the content.

  • “Response to Reviewer”

Following your advice, we have improved the image quality of Figure 2 and provided a clearer caption. These adjustments ensure that the figure is easily interpretable and aligns with the quality standards of our manuscript.

2) “Revised Manuscript Text”

-Improved image quality of Figure 2. & Provided a clearer caption for Figure 2.

  1. Section 5: Conclusion and Discussion:

   - The current combined section is lengthy. Consider splitting it into separate sections: "Discussion" and "Conclusion."

   - Within the "Discussion" section, further organize content by creating subsections. This will enhance clarity and facilitate focused discussions.

  • “Response to Reviewer”

In response to your feedback, we have divided the combined "Conclusion and Discussion" section into two distinct sections: "Discussion" and "Conclusion." We have also organized the "Discussion" section into subsections to clarify and focus the content, enhancing the overall structure and readability.

2) “Revised Manuscript Text”

- Split the "Conclusion and Discussion" section into separate "Discussion" and "Conclusion" sections.

- Created subsections within the "Discussion" section for better organization and focus.

  1. Section 6: Limitations and Future Research Suggestions:

   - Rather than having a standalone section, integrate "Limitations" and "Future Research Suggestions" as subsections within the "Discussion" section.

   - Additionally, consider breaking down the content within this section to improve readability.

  • “Response to Reviewer”

In response to your suggestion, we endeavored to segregate "Limitations" and "Future Research Suggestions." However, after considering the common feedback from other reviewers, we determined that these components were more cohesively presented in an integrated manner. Many reviewers advised that the limitations of our study be directly linked to areas for future enhancement, as they saw a natural connection between the two. Despite this integration, we understand and value your perspective on the importance of readability and agree that a more distinct separation of these sections could enhance the clarity of our manuscript. We will consider this in our approach to future research publications.
